# Development and Evaluation of a Classified and Tailored Community-Based Exercise Program According to the Mobility Level of People with Stroke Using the Knowledge to Action Framework

**DOI:** 10.3390/ijerph17249364

**Published:** 2020-12-14

**Authors:** Minyoung Lee, Seon-Deok Eun, Tae Hyun Cho, Young-Il Shin, Jiyeon Song, Seung Hee Ho

**Affiliations:** 1Department of Healthcare and Public Health Research, National Rehabilitation Research Institute, Seoul 01022, Korea; interlaw88@korea.kr (M.L.); the529@korea.kr (T.H.C.); 2Department of Clinical Research for Rehabilitation, National Rehabilitation Research Institute, Seoul 01022, Korea; esd7786@korea.kr; 3Department of Prosthetics and Orthotics, Korea National College of Welfare, Pyeongtaek 17738, Korea; syi3301@hanmail.net; 4Seoul Junggu Community Health Center, Seoul 04611, Korea; hisong@junggu.seoul.kr

**Keywords:** stroke, community-based exercise program, knowledge to action framework

## Abstract

Attempts to classify the mobility levels of people with stroke (PWS) for a tailored exercise program in community settings have been few. We developed and evaluated a classified and tailored community-based (CTC) exercise program according to the mobility level of PWS. Forty-two PWS were classified into the Supine and Sitting, Sitting and Standing, and Standing and Gait groups, based on a newly developed classification model and participated in a group-based CTC exercise program for 1 h/day twice/week for 10 weeks. The health outcome measures were blood pressure, lipid profile, glucose control, body composition, ventilatory capacity, and physical and psychological function. The rate of agreement on classification results among the physiotherapists was analysed. For all participants, significant improvements were noted in the blood pressure, lipid profile, body composition, ventilatory capacity, and physical and psychological function. The lower mobility groups showed significant improvements in a greater number of health outcomes than the higher mobility group. The physiotherapists’ agreement regarding the classification results was 91.16 ± 5.14%, verifying the model’s possible high relevance to the community. The effective improvement in participant health implied that the CTC exercise program was well tailored to the participants’ mobility levels, particularly the lower mobility groups.

## 1. Introduction

People with stroke (PWS) comprise the third highest number of hospital visits and annual average medical expenses among disabled populations in South Korea [1]. The medical expenses of PWS are approximately six times higher than those of the general population [1]. Further, over 25% of people with a history of stroke are at an increased risk of stroke recurrence [2]. Given the sharp increase in life expectancy in South Korea, this trend is assumed to worsen further. Thus, practical interventions that prevent stroke recurrence and cardiovascular disease and thereby reduce medical care utilisation and expenditures are urgently needed.

Stroke recurrence is influenced by a number of metabolic risk factors, including hypertension, impaired glucose control, dyslipidaemia, obesity, and low cardiorespiratory fitness [3,4]. Exercise is an inexpensive, safe, and effective method of improving metabolic risk factors with minimal side effects in healthy populations [5,6] as well as PWS in community settings [7,8]. Moreover, exercise reportedly improves physical function in terms of balance, strength, endurance, and walking performance [8,9,10,11,12,13] as well as psychological function in PWS [14].

However, the effective community-based exercise programs for PWS in previous studies mostly targeted people who were able to walk independently [8,10,11,12,13,14], meaning that evidence regarding those who cannot walk or sit independently are lacking. Even among PWS who are able to walk independently, mobility levels vary according to the degree of assistance needed despite participation in the same intervention. Limited targeted exercise programs might discourage PWS from participating in exercise, leading them to secondary health complications and further disabling conditions. This phenomenon highlights the need for a classified and tailored community-based (CTC) exercise program that involves a wider stroke population with a variety of mobility levels.

In developing a CTC exercise for PWS, prior evidence is not available for either community or clinical settings. In the clinical setting, there are personnel resources with highly specialised knowledge required to perform various clinical and physical examinations and provide tailored interventions on a one-to-one basis, while in community settings, health professionals with varied work experience have insufficient time to perform examinations of and provide interventions to individuals due to uncontrolled conditions such as manpower or budgetary shortages. 

The knowledge translation method might be useful for solving the problem of the gap between prior evidence and the real world, which is a dynamic and iterative process of moving knowledge into action that includes the synthesis, dissemination, exchange, and ethically sound application of knowledge to provide more effective health services and strengthen the health care system [15]. Graham and colleagues suggested a theoretical framework for the knowledge translation method, called the knowledge to action framework, that aimed to convert knowledge into action through several phases [16]: (1) identify the problem; (2) identify, review, and select the relevant knowledge; (3) assess barriers to the local use of knowledge; (4) tailor and implement a program; (5) monitor and evaluate program progress; and (6) sustain the knowledge. 

The purpose of this study was to develop and evaluate a CTC exercise program according to the mobility levels of PWS using the knowledge to action framework. In the evaluation, first, the effectiveness of the program on health outcomes was analysed; second, the rate of agreement among the physiotherapists regarding the classification results was analysed; and third, the physiotherapists’ experience and perceived supplemental points of the CTC exercise program implementation were explored.

## 2. Materials and Methods

### 2.1. Community Partner

We established community-academic partnerships with eight community physiotherapists in the community health centres of three different districts of Seoul, i.e., Jung-gu, Dongjak-gu, and Jongno-gu, to enhance the relevance of the research results in the community context. We built an equitable collaboration while developing and evaluating a CTC exercise program. Three physiotherapists were female. The physiotherapists were aged 30–48 years, with a clinical experience of 5–20 years.

### 2.2. Target Population

The research team advertised information about the study on a message board at the community health centre for 1 month to recruit PWS. A total of 50 volunteers were assessed for eligibility; of them, eight were excluded from the study for the following reasons: ineligible according to the inclusion criteria (n = 5) and too busy to participate (n = 3), leaving 42 enrolled participants. PWS were included if they had experienced a stroke ≥1 year prior and were not already engaging in regular exercise (≥3 times/week, moderate intensity). They were excluded if they were unable to communicate sufficiently to participate in the assessment and intervention phases (Mini-Mental State Examination score <18) [17], presented with uncontrolled hypertension or cardiac conditions, or were receiving ongoing inpatient rehabilitation. This study was approved by the Institutional Review Board of the Korea National Rehabilitation Center (NRC-2018–01–008). Written informed consent was obtained from all participants, and the study conformed to the Declaration of Helsinki guidelines.

### 2.3. Procedure

All phases of the study were based on the knowledge to action framework [16] (Table 1). The researchers and physiotherapists met regularly once a week for 2 months to design the CTC exercise program. Literature reviews, expert consultations, and focus group interviews were conducted as needed in each phase. 

#### 2.3.1. Identify the Problem 

In the first regular meeting, the researchers and physiotherapists identified the problems related to implementing an exercise program for the PWS in a community health centre. The following two problems were identified: (1) difficulty in implementing a tailored exercise program due to a lack of a standardised method or criteria for judging participant mobility level and (2) tendency for the exercise program to have a limited ability to target PWS roughly judged as those who could walk independently because they were easy to control in groups to minimise the risk of falls.

#### 2.3.2. Identify, Review, and Select Knowledge to Adapt to Community Context

Literature review. In previous studies, community-based exercise programs for PWS consisted of some combination of postural control, stretching, muscle strength and resistance exercises, and aerobic endurance and mostly targeted PWS who were able to walk independently with or without an assistive device [8,10,11,12,13,14]. Various health outcome measures were adopted to examine the effectiveness of those programs at improving metabolic risk factors and physical and psychological function [8,10,11,12,13,14]; however, few attempts were made to classify the mobility levels of PWS to implement a tailored exercise program.Expert consultation. For uncovered knowledge through literature review, we were consulted three times by professors and practitioners in the fields of sports and physiotherapy. To determine participant mobility level, sub-categories of the mobility chapter of the International Classification of Functioning, Disability, and Health framework [18], such as changing and maintaining body position (d410-d429) and walking (d450), were adopted. To measure the mobility level based on those evaluation criteria, a combination of the following three tests were recommended: (1) the Motor Assessment Scale (MAS) item 4 (Sit-to-Stand), (2) the 30-sec Chair Sit-and-Stand test, and (3) the 8-foot up-and-go test. These tests are easy to perform in the community, do not require special equipment, and can be performed quickly within a limited physical space. Their validity and reliability have been documented [19,20]. Upon combination of the three tests, the experts suggested that the participants be divided into the following three groups: (1) Supine and Sitting, (2) Sitting and Standing, and (3) Standing and Gait.Development of a classification model. Given the combination of available prior evidence, expert consultation, and field experience of physiotherapists, we designed a classification model according to the mobility level (Table 2). The criteria for each mobility level were determined conservatively with reference to the reported values in previous studies [21] and accounted for the risk of falls that could occur if the participants conducted the exercise unsupervised. For example, to belong to the Standing and Gait group, the participant should satisfy these three criteria: (1) MAS item 4 score > 3, (2) ability to perform eight sit-and-stand repetitions within 30 s, and (3) ability to perform the 8-foot up-and-go test in less than 9 sec.Development of a tailored exercise program. The exercise program for the Standing and Gait group was adapted from the previously suggested community-based exercise program for PWS [8,11,13], while those for the Supine and Sitting and Sitting and Standing groups were newly developed by the research team by modification of the standing position for the Standing and Gait group into supine and sitting positions if possible or modification of the patterns of physical rehabilitation, such as the proprioceptive neuromuscular facilitation method [22,23]. The exercise components of a CTC exercise program consisted of stretching and postural control, functional strengthening, and agility and fitness [8,10,11,12,13,14] (Appendix A
Table A1).

#### 2.3.3. Assess Barriers to Local Use of Knowledge

The physiotherapists identified several contextual barriers to knowledge use in the community, including: (1) shortage of time and manpower to provide personalised one-to-one exercise training and (2) insufficient knowledge and lack of experience on the part of the physiotherapists to provide a tailored exercise program for PWS with low mobility.

#### 2.3.4. Tailor and Implement the Program

To address the first identified problem, a CTC exercise program was planned that was to be administered groups of PWS with the same mobility level by an instructing physiotherapist and one or two assistant physiotherapists. The group-based CTC exercise program was conducted for 1 h/day twice/week for 10 weeks at the community health centre. In addition, an exercise manual detailing the exercise motions and an exercise diary book, including individual goal setting, pre-intervention test results, and a daily exercise log, were provided to participants to encourage them to perform the exercises at home during the intervention period. Before starting each exercise session, the physiotherapists briefly assessed whether the participants had performed the exercises at home and if they were encountering obstacles. At the end of every exercise session, the physiotherapists selected some motions in the exercise manual tailored to each participant’s mobility level and assigned them as homework for the week. 

To address the second identified problem, after the CTC exercise program was designed, three workshops were held for the physiotherapists with the specialised experts to promote co-learning and standardisation of the CTC exercise program delivery according to mobility level and review any precautions requiring attention.

#### 2.3.5. Monitor and Evaluate

The health outcomes were evaluated before and after the intervention. After the intervention period, the rate of agreement among the physiotherapists regarding the classification results was analysed to validate the ability of the developed classification model to appropriately reflect the participants’ mobility levels. The physiotherapists’ experience and perceived supplemental points during the implementation of the CTC exercise program’s implementation were gathered from physiotherapist intervention logs at every session.

### 2.4. Data Collection

#### 2.4.1. Participants’ Characteristics

The participants’ baseline characteristics (age, sex, duration since stroke, cognitive status, severity of disability, and mobility level classification) were obtained. Cognitive status was based on the Mini-Mental State Examination score [17]. Severity of disability was based on the authorised disability ratings for individuals with neurological disorders defined by the Ministry of Health and Welfare, South Korea: 1, need for total assistance with gait and performing activities of daily living; 2, need for great assistance; 3, need for partial assistance; 4, need for intermittent assistance; 5, partially independent; and 6, totally independent. The mobility level classification was based on the currently developed classification model.

#### 2.4.2. Health Outcome Measures

For health outcomes, the following metabolic risk factors and physical and psychological function were measured.

Resting blood pressure. Systolic blood pressure (SBP) and diastolic blood pressure (DBP) were recorded using a semiautomated sphygmomanometer after a 10-min seated rest [24].Lipid profile and glucose control. After an overnight fast, blood was collected from each participant and total cholesterol (Total-C), high-density lipoprotein cholesterol (HDL-C), low-density lipoprotein cholesterol (LDL-C), triglyceride (TG), and haemoglobin A1c (HbA1c) levels were measured [25,26].Body composition. Body mass index and waist circumference were measured [27,28].Ventilatory capacity. Forced expiratory volume in 1 s (FEV1), forced vital capacity (FVC), and the FEV1/FVC ratio were assessed [29].Physical function. MAS item 4 score (range, 0–5, with higher scores indicating better mobility) (Lannin, 2004), upper extremity portion of the Fugl–Meyer Assessment (UE-FMA) (range, 0–66, with higher scores indicating better upper extremity function) [30], the Modified Barthel Index (MBI) (range, 0–100, with higher scores indicating better ability to perform basic activities of daily living) [31], the 30-s chair-to-stand test score, and the 8-foot up-and-go test score among the Senior Fitness Test subsets [19] were assessed.Psychological function. The Patient Health Questionnaire (PHQ)-9 (range, 0–27, with higher scores indicating more severe depression) [32].

#### 2.4.3. Physiotherapists’ Agreement with Classification Results

Each physiotherapist determined whether the classification results appropriately reflected the participants’ mobility levels. For the decision rule, “Complete” was adopted [33], which was operationalized as the proportion of judges that rated a classification result as completely representative of the mobility level.

#### 2.4.4. Physiotherapist Experience and Perceived Supplemental Points 

The physiotherapists briefly recorded their experience and perceived supplement points during CTC exercise program implementation in the field notes at every exercise session.

### 2.5. Data Analysis

Health outcomes were analysed all participants and each group using SPSS 21.0.b (IBM SPSS Inc., Chicago, IL, USA). The normality of the distribution was assessed using the Kolmogorov–Smirnov test. For all participants, the paired t-test was performed to compare all of the pre- and post-intervention values except MBI, MAS item 4, and FMA, which were analysed using the Wilcoxon signed-rank test because they were non-normally distributed. For each group, the Wilcoxon signed-rank test was performed to compare all of the pre- and post-intervention values. 

A summative content analysis was performed to evaluate the physiotherapists’ responses by counting and comparing keywords in the text with the purpose of understanding the contextual meaning of the sentence [34]. Two researchers independently coded the transcribed interviews and determined the themes based on the keywords that were presented in the codes. Those themes were discussed among the researchers and physiotherapists to verify their accuracy and representativeness.

## 3. Results

### 3.1. Participants’ Characteristics

Forty-two participants completed all of the assessments. The mean participant age was 64.79 ± 10.54 years; 54.76% of the participants were female. The mean duration since stroke was 9.54 ± 5.79 years. Stroke severity varied from grade 1 to grade 6. At baseline, the largest number of participants was assigned to the Sitting and Standing group (45.24%), followed by the Supine and Sitting (35.71%) and Standing and Gait (19.05%) groups. The participants’ characteristics are shown in Table 3.

### 3.2. Health Outcomes

For all participants, significant improvements were noted in blood pressure, lipid profile, body composition, ventilatory capacity, and physical and psychological function (Table 4). The lower mobility groups (Supine and Sitting, Sitting, and Standing) showed significant improvements in more health outcome measures than did the higher mobility group (Standing and Gait) (Table 5).

Blood pressure. For all participants, significant increases were noted in SBP (*p* = 0.028) and DBP (*p* < 0.001). For the Supine and Sitting group, significant increases in DBP (*p* = 0.007) were observed. For the Sitting and Standing group, significant increases in SBP (*p* = 0.014) and DBP (*p* = 0.002) were seen. For the Standing and Gait group, significant increases in DBP (*p* = 0.017) were observed.Lipid profile and glucose control. For all participants, significant increases in Total-C (*p* = 0.013) and LDL-C (*p* < 0.001) were observed. For the Supine and Sitting group, significant increases in Total-C (*p* = 0.005) and LDL-C (*p* = 0.004) were noted. For the Sitting and Standing group, significant increases in LDL-C (*p* = 0.022) were seen.Body composition. For all participants, significant increases in waist circumference (*p* = 0.011) were noted.Ventilatory capacity. For all participants, significant increases in FEV (*p* = 0.005) and FVC (*p* = 0.018) were seen. For the Sitting and Standing group, significant increases in FVC (*p* = 0.045) were observed. For the Standing and Gait group, significant increases in FEV (*p* = 0.025) were noted.Physical function. For the total participants, significant increases in the UE-FMA (*p* < 0.001), MBI (*p* = 0.009), 30-sec chair-to-stand test (*p* = 0.001), and 8-foot up-and-go test (*p* = 0.011) results were seen. For the Supine and Sitting groups, significant increases in MAS item 4 (*p* = 0.041), UE-FMA (*p* = 0.004), and MBI (*p* = 0.003) were observed. For the Sitting and Standing group, significant increases in the UE-FMA (*p* = 0.019), 30-sec chair-to-stand test (*p* = 0.006), and 8-foot up-and-go test (*p* = 0.005) results were noted.Psychological function. For all participants, significant increases in the PHQ-9 (*p* = 0.001) score were seen. For the Supine and Sitting group, significant increases in the PHQ-9 (*p* = 0.006) score were observed.

### 3.3. Physiotherapists’ Agreement with Classification Results

Three physiotherapists belonging to the community health centre of the Jung-gu district judged the 15 participants in their charge, two physiotherapists belonging to the Dongjak-gu district judged the eight participants in their charge, and three physiotherapists belonging to the Jongno-gu district judged the 19 participants in their charge. The mean rate of agreement of all of eight physiotherapists was 91.16 ± 5.14%.

### 3.4. Physiotherapist Experience and Perceived Supplemental Points

The experiences and perceived supplemental points of the physiotherapists addressed the following four main themes: (1) classification, (2) group-based intervention, (3) tailored exercise program, and (4) self-exercise at home.

#### 3.4.1. Classification 

The majority of the physiotherapists agreed with the classification results: five physiotherapists reported that some participants who were assigned to the lower mobility groups could actually stand or perform gait to some degree; however, they were at risk of falling and injury due to decreased balance ability. Thus, the physiotherapists agreed that the classification results were appropriate, particularly in terms of self-exercise at home; two physiotherapists recognised that participating in an exercise class of a similar mobility level seemed to increase the participants’ confidence, while they often felt shy when they exercised on a one-to-one basis with a physiotherapist or with PWS of different mobility levels. Besides, four physiotherapists recommended that before the classification was applied, it would be necessary to persuade the participants to accept the result since most of them hoped to be assigned to the high mobility group and were very disappointed when they were assigned to a lower mobility group.

#### 3.4.2. Group-Based Intervention

The majority of the physiotherapists recommended that the exercise classes for the PWS with different mobility levels be offered separately because those in the lower mobility group required more help due to their greater risk of falling; however, if this was impossible due to a lack of space or time, the patients should at least be grouped by ability level. They also suggested that one physiotherapist could instruct four or five participants of the same level at a time. If a greater number of participants required treatment, one or more assistant physiotherapists should be involved to prevent an accident. The physiotherapists also suggested that participants be addressed by name to correct their poor posture since they experienced that this approach highly motivated them to engage in the exercise.

#### 3.4.3. Tailored Exercise Program 

The majority of the physiotherapists reported that recreational programs such as balloon badminton and volleyball were favoured by participants and that various creative rules applied to recreational program were helpful to draw their continued interest, such as matches between two teams or hitting only a ball the colour that was called by the physiotherapist. Besides, two physiotherapists suggested increasing the difficulty of the motions and developing additional exercises for the high mobility group.

#### 3.4.4. Self-Exercise at Home 

The physiotherapists recognised that tailored guidance for self-exercise at home was necessary, as evidenced by the participants’ positive reactions to the provided exercise manual and diary log and recommended that the self-exercise motions should consist of supine and sitting positions to prevent falling accidents at home. Some of the physiotherapists suggested that the contents of the self-exercise program be provided in video format via a mobile application to help the participants follow the motions.

## 4. Discussion

This study is one of the first to develop and evaluate a CTC exercise program tailored to the mobility level of PWS. The participants were classified into three groups accordingly and participated in the group-based exercise program tailored to their mobility level. The classification model and tailored exercise program were developed using the knowledge to action framework, which consisted of several phases to move knowledge into action. As a result, the participants showed significant improvements in metabolic risk factors as well as physical and psychological function. The majority of the physiotherapists agreed with the classification results.

Significant improvements in metabolic risk factors were observed. This is noteworthy since stroke recurrence was influenced by modifiable metabolic risk factors [3,4]. However, few attempts have been made to analyse the efficacy of a community-based exercise program on metabolic risk factors in PWS. In particular, PWS who could not walk or sit independently have not been analysed in previous studies. A previous study [8] that analysed the metabolic effects of a 19-week group-based exercise program in PWS who could walk independently reported significant improvements in DBP, HDL-C, and cardiorespiratory fitness. One systematic review [7] targeted individuals with subacute stroke and transient ischemic attack and reported that exercise improved SBP, fasting glucose, and HDL-C. In the current study, significant improvements in SBP, DBP, Total-C, LDL-C, waist circumference, FEV, and FVC were noted. These results are meaningful in that the effect of a CTC exercise program on metabolic risk factors in chronic PWS was studied in high and low mobility groups. Noticeably, the lower mobility groups showed significant improvements in more variables related to metabolic risk factors than the higher mobility group, implying that tailored exercise programs for the lower mobility groups targeted the appropriate mobility level well.

For physical function, significant improvements in mobility, upper extremity function, and ability to perform activities of daily living were observed. Tests for assessing mobility (i.e., MAS item 4, 30-sec chair-to-stand test, and 8-foot up-and-go test) were adopted as recommended by experts due to the ease of performing them in community settings and the ability to assess PWS with various mobility levels, while many previous studies adopted the Berg Balance Scale [8,10,11], 5- or 10-m walk test [8,12,13], and the 2- or 6-min walk test [8,11,12,13] to assess balance and walking capacity and speed of PWS in community settings. However, these previously adopted tests took relatively much time to assess considering the local context of the community health canter of South Korea, and low mobility groups would be excluded at all from assessing walking capacity and speed. This situation highlights the need to discover or develop appropriate methods for assessing PWS with various mobility levels with high sensitivity considering the community context.

For physical function, the Supine and Sitting group showed improvements in basic physical function, while the Sitting and Standing group improvements in lower limb strength and dynamic balance, implying that the exercise programs for the lower mobility groups targeted their mobility levels well. In contrast, the Standing and Gait group showed no significant improvement in physical function, possibly since it included the smallest number of participants among the three groups or the program failed to target the participants’ mobility levels well. The physiotherapists reported via field notes that increases in the difficulty of the motions and the development of additional exercise contents would be necessary for the high mobility group. For the Standing and Gait group, it is worthy referring to Olafsdottir and colleagues’ recent report about the feasibility of ActivABLES to promote home-based exercise and physical activity of community-dwelling stroke survivors, which consisted of six interactive tangible tools [35]. 

The physiotherapists presented their opinions based on their experiences and perceived supplemental points. For example, most of them agreed with the classification results. In particular, the physiotherapists’ judgments considered that the participants conducted self-exercise themselves at home without supervision, a consideration of ours in the mobility level determination criteria. Regarding the group-based intervention, the physiotherapists suggested that one physiotherapist could instruct up to five participants at a time. In some previous studies, one class included up to six or nine participants with a single physiotherapist and/or one or two assistant physiotherapists [10,13]. Given that the exercise program of the current study included PWS with a low mobility level, this suggestion seems reasonable.

In the current study, the knowledge to action framework enabled us to identify and solve problems in the community setting and develop a CTC exercise program with high relevance by establishing community–academic partnerships. Further, the physiotherapists obtained local knowledge based on their experiences by evaluating this CTC exercise program. Importantly, this local knowledge would promote their empowerment, enabling them to sustain health services that consider the local context even after study completion.

This study has several limitations. First, it lacked a control group, which limits its power. Thus, future large-scale randomised trials are needed to definitely conclude the effectiveness of this CTC exercise program. Second, the intervention was limited to 10 weeks. Therefore, a long-term study accompanied by follow-up tests is recommended to verify the longer-term effect of the program on health outcomes. Third, it was conducted using a small sample size, in particular in the Standing and Gait group, limiting the generalizability of the result of current study.

## 5. Conclusions

The group-based CTC exercise program effectively improved the health outcomes of PWS, such as metabolic risk factors and physical and psychological function. Further, the majority of the community physiotherapists agreed that the classification results reflected the mobility levels of the PWS using this newly developed classification model, accompanied by positive experiences during CTC exercise program implementation. These results imply that the classification model and a CTC exercise program tailored to the mobility level of each study group would be effective and feasible in the community setting.

## Figures and Tables

**Table 1 ijerph-17-09364-t001:** Knowledge to action framework.

Phase	Tools	Products
Identify the problem	Regular meeting	**Identified problems** Difficulty in implementing a tailored exercise program due to the lack of a standardised method or criteria for assessing mobility level Limited ability to target the PWS who could walk independently
2. Identify, review, and select knowledge	Literature review Expert consultation Regular meeting	**Selected knowledge** Evidence about the components and contents, evaluation methods, and effectiveness of a community-based exercise program for the PWS Evaluation criteria for determining mobility level and method for measuring it Classification model and corresponding exercise program
3. Assess barriers to local use of knowledge	Regular meeting	**Assessed barriers** Shortage of time and manpower to provide personalised one-to-one exercise training Insufficient knowledge and lack of experience of community PTs to provide a tailored exercise program for the PWS with a low mobility level
4. Tailor and implement	Regular meeting Workshop Pilot study	**CTC exercise program and workshop to promote co-learning** Group-based CTC exercise program according to mobility level Workshops to promote co-learning and standardise delivery of the CTC exercise program to PWS
5. Monitor and evaluate	Quantitative evaluation Qualitative evaluation	**Quantitative and qualitative findings** Assessment of health outcomes Exploration of responses of the community PTs to CTC exercise program implementation Rate of community PT agreement with classification results

PWS, people with stroke; PTs, physiotherapists; CTC, classified and tailored community-based.

**Table 2 ijerph-17-09364-t002:** Classification model according to the mobility level.

Group	MAS Item 4 Score	Condition	30-s Chair Sit-and-Stand Test	Condition	8-Foot Up-and-Go Test
Supine and Sitting	<4 points	OR	<6 times		
Sitting and Standing	>4 points but <6 points	AND	>6 times but <8 times		
Standing and Gait	6 points	AND	>8 times	AND	<9 s

MAS, Motor Assessment Scale.

**Table 3 ijerph-17-09364-t003:** Participants’ characteristics (*N* = 42).

Characteristic	Value
Age, years	64.79 ± 10.54
Sex, female/male	23/19 (54.76/45.24)
Duration since stroke, years	9.54 ± 5.79
Cognitive status, MMSE score	25.40 ± 3.96
Stroke severity ^a^	
Grade 1	1 (2.38)
Grade 2	10 (23.81)
Grade 3	10 (23.81)
Grade 4	13 (30.95)
Grade 5	6 (14.29)
Grade 6	2 (4.76)
Group classification by mobility level	
Supine and Sitting	15 (35.71)
Sitting and Standing	19 (45.24)
Standing and Gait	8 (19.05)

Values are shown as mean ± SD or *n* (%). MMSE, Mini-Mental State Examination. ^a^ Lower grade reflects a more severe state of disability.

**Table 4 ijerph-17-09364-t004:** Baseline scores and change in scores for the health outcomes of all the participants.

Measurement		Pre-Intervention	Post-Intervention	Gap	*p*-Value
Blood pressure	SBP, mmHg	135.45 ± 19.12	129 ± 17.53	−6.45 ± 18.31	0.028 ^a^*
	DBP, mmHg	77.90 ± 9.63	71.86 ± 8.71	−6.05 ± 6.72	0.000 ^a^*
Lipid profile, glucose control	Total-C, mmol/L	174.10 ± 40.43	163.62 ± 36.50	−10.48 ± 26.09	0.013 ^a^*
	HDL-C, mmol/L	50.21 ± 12.16	48.36 ± 13.18	−1.85 ± 6.94	0.092 ^a^
	LDL-C, mmol/L	108.31 ± 37.46	94.71 ± 31.63	−13.60 ± 23.02	0.000 ^a^*
	TG, mg/dL	124.71 ± 44.74	132.31 ± 53.73	7.60 ± 53.69	0.365 ^a^
	HbA1c, %	5.83 ± 0.88	5.84 ± 1.01	0.01 ± 0.59	0.938 ^a^
Body composition	BMI, kg/m^2^	24.85 ± 3.99	24.31 ± 4.04	−0.54 ± 1.92	0.080 ^a^
	Waist circumference, cm	87.27 ± 10.84	85.50 ± 9.17	−1.77 ± 4.31	0.011 ^a^*
Ventilatory capacity	FEV, L	1.87 ± 0.56	2.09 ± 0.64	0.22 ± 0.48	0.005 ^a^*
	FVC, L	2.62 ± 0.78	2.79 ± 0.75	0.17 ± 0.45	0.018 ^a^*
	FEV_1_/FVC, %	69.57 ± 12.87	70.78 ± 15.73	1.20 ± 16.60	0.641 ^a^
Physical function	MAS item 4 score	4.90 ± 1.76	5.10 ± 1.48	0.19 ± 1.38	0.377 ^a^
	UE-FMA score	45.17 ± 22.57	50.26 ± 20.20	5.10 ± 6.85	0.000 ^b^*
	MBI score	84.79 ± 16.16	90.26 ± 10.31	5.48 ± 12.97	0.009 ^b^*
	Chair-to-stand, count	9.92 ± 4.12	11.93 ± 4.43	1.96 ± 2.94	0.001 ^a^*
	8-foot up-and-go, seconds	16.04 ± 8.47	13.32 ± 6.82	−1.91 ± 3.78	0.011 ^a^*
Psychological function	PHQ-9 score	9.33 ± 5.75	6.31 ± 5.06	−3.02 ± 5.47	0.001 ^a^*

SBP, systolic blood pressure; DBP, diastolic blood pressure; Total-C, total cholesterol; HDL-C, high-density lipoprotein cholesterol; LDL-C, low-density lipoprotein cholesterol; TG, triglyceride; HbA1c, haemoglobin A1c; BMI, body mass index; FEV, forced expiratory volume; FVC, forced vital capacity; MAS, Motor Assessment Scale; UE-FMA, upper extremity portion of the Fugl–Meyer Assessment; MBI, Modified Barthel Index; PHQ-9, Patient Health Questionnaire-9; * *p* < 0.05; ^a^ Analysed using the paired *t*-test; ^b^ Analysed using the Wilcoxon signed-rank test.

**Table 5 ijerph-17-09364-t005:** Baseline score and change in scores of the health outcomes according to the groups.

Measurement		Pre-Intervention	Post-Intervention	Gap	*p*-Value
Blood pressure	SBP, mmHg				
	Supine and Sitting	137.47 ± 10.76	135.93 ± 20.2	−1.53 ± 19.64	0.513
	Sitting and Standing	135.21 ± 24.95	124.32 ± 15.34	−10.89 ± 18.96	0.014 *
	Standing and Gait	132.25 ± 17.00	127.13 ± 14.56	−5.13 ± 12.7	0.310
	DBP, mmHg				
	Supine and Sitting	79.4 ± 7.65	73.53 ± 8.58	−5.87 ± 6.45	0.007 *
	Sitting and Standing	75.79 ± 9.23	69.84 ± 9.04	−5.95 ± 6.75	0.002 *
	Standing and Gait	80.13 ± 13.60	73.5 ± 8.19	−6.63 ± 7.98	0.017 *
Lipid profile, glucose control	Total-C, mmol/L				
	Supine and Sitting	189.13 ± 46.89	167.80 ± 43.79	−21.33 ± 31.14	0.005 *
	Sitting and Standing	161.05 ± 33.62	159.79 ± 33.85	−1.26 ± 21.78	0.732
	Standing and Gait	176.88 ± 36.23	164.88 ± 30.68	−12.00 ± 18.49	0.141
	HDL-C, mmol/L				
	Supine and Sitting	49.26 ± 13.34	46.46 ± 16.92	−2.80 ± 6.71	0.147
	Sitting and Standing	49.96 ± 11.96	49.64 ± 10.75	−0.31 ± 7.01	0.647
	Standing and Gait	52.58 ± 11.56	48.88 ± 11.63	−3.7 ± 7.28	0.161
	LDL-C, mmol/L				
	Supine and Sitting	122.55 ± 44.64	98.79 ± 38.75	−23.76 ± 27.71	0.004 *
	Sitting and Standing	98.13 ± 31.59	91.69 ± 29.51	−6.43 ± 16.63	0.022 *
	Standing and Gait	105.79 ± 30.73	94.20 ± 23.63	−11.60 ± 22.16	0.161
	TG, mg/dL				
	Supine and Sitting	141.27 ± 47.69	145.73 ± 40.13	4.47 ± 54.36	0.865
	Sitting and Standing	111.95 ± 40.21	121.68 ± 55.88	9.74 ± 42.02	0.409
	Standing and Gait	124.00 ± 44.59	132.38 ± 70.45	8.38 ± 80.02	0.779
	HbA1c, %				
	Supine and Sitting	5.88 ± 0.90	5.95 ± 0.71	0.07 ± 0.42	0.362
	Sitting and Standing	5.77 ± 0.90	5.77 ± 1.25	0.00 ± 0.77	0.195
	Standing and Gait	5.89 ± 0.90	5.80 ± 0.93	−0.09 ± 0.38	0.551
Body composition	BMI, kg/m^2^				
	Supine and Sitting	24.45 ± 3.42	23.75 ± 3.02	−0.7 ± 1.34	0.103
	Sitting and Standing	25.72 ± 4.58	25.41 ± 5.04	−0.31 ± 2.49	0.205
	Standing and Gait	23.53 ± 3.38	22.75 ± 2.32	−0.78 ± 1.33	0.123
	Waist circumference, cm				
	Supine and Sitting	87.97 ± 9.10	84.90 ± 7.75	−3.07 ± 5.70	0.084
	Sitting and Standing	88.26 ± 11.78	87.61 ± 10.16	−0.66 ± 2.75	0.337
	Standing and Gait	83.63 ± 12.11	81.63 ± 8.73	−2 ± 4.17	0.205
Ventilatory capacity	FEV_1_, L				
	Supine and Sitting	1.85 ± 0.51	1.98 ± 0.55	0.13 ± 0.32	0.173
	Sitting and Standing	1.83 ± 0.63	2.05 ± 0.77	0.22 ± 0.60	0.112
	Standing and Gait	2.00 ± 0.52	2.39 ± 0.36	0.39 ± 0.41	0.025 *
	FVC, L				
	Supine and Sitting	2.52 ± 0.49	2.61 ± 0.57	0.09 ± 0.28	0.280
	Sitting and Standing	2.63 ± 1.03	2.85 ± 0.90	0.23 ± 0.61	0.045 *
	Standing and Gait	2.80 ± 0.54	3.00 ± 0.68	0.20 ± 0.26	0.063
	FEV_1_/FVC, %				
	Supine and Sitting	71.80 ± 11.47	70.17 ± 19.55	−1.63 ± 21.18	0.470
	Sitting and Standing	67.21 ± 13.79	68.58 ± 14.81	1.37 ± 14.30	0.513
	Standing and Gait	71.00 ± 13.74	77.13 ± 7.83	6.13 ± 12.11	0.128
Physical function	MAS item 4 score				
	Supine and Sitting	3.00 ± 1.73	4.00 ± 1.77	1.00 ± 1.81	0.041 *
	Sitting and Standing	5.95 ± 0.23	5.58 ± 0.96	−0.37 ± 0.96	0.059
	Standing and Gait	6.00 ± 0.00	6.00 ± 0.00	0.00 ± 0.00	1.000
	UE-FMA score				
	Supine and Sitting	37.73 ± 25.16	45.80 ± 21.34	8.07 ± 7.01	0.004 *
	Sitting and Standing	45.05 ± 22.27	48.63 ± 21.70	3.58 ± 6.95	0.019 *
	Standing and Gait	59.38 ± 9.66	62.50 ± 6.82	3.13 ± 4.76	0.109
	MBI score				
	Supine and Sitting	72.40 ± 16.32	84.73 ± 11.13	12.33 ± 11.71	0.003 *
	Sitting and Standing	88.74 ± 12.56	92.00 ± 9.02	3.26 ± 13.84	0.147
	Standing and Gait	98.63 ± 1.19	96.50 ± 6.82	−2.13 ± 6.20	0.480
	Chair-to-Stand, count				
	Supine and Sitting	3.50 ± 2.38	6.95 ± 2.41	2.44 ± 3.08	0.068
	Sitting and Standing	9.97 ± 2.3	12.45 ± 3.51	2.33 ± 2.96	0.006 *
	Standing and Gait	13.00 ± 4.55	14.00 ± 5.05	1.00 ± 2.98	0.575
	8-foot up-and-go, seconds				
	Supine and Sitting	21.49 ± 7.7	17.45 ± 8.43	−4.14 ± 8.11	0.225
	Sitting and Standing	17.76 ± 8.34	14.51 ± 6.34	−1.95 ± 2.42	0.005 *
	Standing and Gait	8.09 ± 1.02	7.67 ± 1.5	−0.42 ± 0.94	0.208
Psychological function	PHQ-9 score				
	Supine and Sitting	10.93 ± 6.03	6.20 ± 5.58	−4.73 ± 5.36	0.006 *
	Sitting and Standing	9.16 ± 5.87	7.26 ± 5.04	−1.89 ± 5.61	0.176
	Standing and Gait	6.75 ± 4.4	4.25 ± 3.85	−2.5 ± 5.15	0.176

SBP, systolic blood pressure; DBP, diastolic blood pressure; Total-C, total cholesterol; HDL-C, high-density lipoprotein cholesterol; LDL-C, low-density lipoprotein cholesterol; TG, triglyceride; HbA1c, haemoglobin A1c; BMI, body mass index; FEV, forced expiratory volume; FVC, forced vital capacity; MAS, Motor Assessment Scale; UE-FMA, upper extremity portion of the Fugl–Meyer Assessment; MBI, Modified Barthel Index; PHQ-9, Patient Health Questionnaire-9. * *p* < 0.05.

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
