# Peer review of "Development and Evaluation of a Classified and Tailored Community-Based Exercise Program According to the Mobility Level of People with Stroke Using the Knowledge to Action Framework"

_ijerph, 2020, doi:10.3390/ijerph17249364_

Round 1

Reviewer 1 Report

It is a well written paper with important implications in community care. I would suggest one minor issue to address before possible publication; the authors state that "eight were excluded from the study for various reasons", more details can be provided. 

Reviewer 2 Report

The main take-away on this paper is that there are many outcomes with few participants in each groups. The authors should discuss about this limits in the discussion. Having 8 participants in one group does make it generalizable to this population (Standing & Gait). 

I feel that the intervention was not able to improve physical function in the Standing & Gait. Could the authors comment on this? In table 5, only two outcomes improved in this group, Diastolic blood pressure (which decreased, but is clinically insignificant because it stayed in the normal range) and FEV. I feel that the intervention was not adequate enough to improve physical function in this group, compared to the other groups (supine & sitting and sitting and standing). The authors should comment on that in their discussion.

The authors report that they measured cardiorespiratory fitness (CRF) with FEV and FVC. They may be associated with CRF, but these are ventilatory parameters and not associated with cardiovascular fitness. I recommend that the authors change this in their paper.

Overall, I believe it is a good paper but it has its limits in generalizability because of a very low statistical power and this should be discussed in the paper.

Reviewer 3 Report

The authors have done a good job of research that will undoubtedly contribute to clinicians who work with stroke patients. As a physiotherapist I have really enjoyed reading it. The manuscript is well structured and easy to read so that readers interested in the subject will be able to follow and adequately replicate the procedure carried out. The results are generalizable and can be contextualized in other countries with minimal changes. As a minor change:
- I would ask the authors to delete the last sentence of the "Community partner" section (lines 84-86).
- Please,
introduce a reference from 2020 to update the bibliography as much as possible, since there are important works related to the subject of the study in recent months.
This is all. Regards
